# In-situ formation of one-dimensional coordination polymers in molecular junctions

Anton Vladyka [1,2], Mickael L. Perrin[2], Jan Overbeck[1,2,3], Rubén R. Ferradás[4,5], Víctor García-Suárez[4,6], Markus Gantenbein[7], Jan Brunner[1], Marcel Mayor [7,8,9], Jaime Ferrer [4,6] & Michel Calame [1,2,3]

We demonstrate the bottom-up in-situ formation of organometallic oligomer chains at the single-molecule level. The chains are formed using the mechanically controllable break junction technique operated in a liquid environment, and consist of alternating isocyano-terminated benzene monomers coordinated to gold atoms. We show that the chaining process is critically determined by the surface density of molecules. In particular, we demonstrate that by reducing the local supply of molecules within the junction, either by lowering the molecular concentration or by adding side groups, the oligomerization process can be suppressed. Our experimental results are supported by ab-initio simulations, confirming that the isocyano terminating groups display a high tendency to form molecular chains, as a result of their high affinity for gold. Our findings open the road for the controlled formation of one-dimensional, single coordination-polymer chains as promising model systems of organometallic frameworks.

[1] Department of Physics, University of Basel, Klingelbergstrasse 82, CH-4056 Basel, Switzerland. [2] Empa, Swiss Federal Laboratories for Materials Science and Technology, Überlandstrasse 129, CH-8600 Dübendorf, Switzerland. [3] Swiss Nanoscience Institute, Klingelbergstrasse 82, CH-4056 Basel, Switzerland. [4] Department of Physics, University of Oviedo, 33007 Oviedo, Spain. [5] Laboratoire de Chimie et Physique Quantiques, IRSAMC, Université Toulouse III - Paul Sabatier, CNRS, 118 Route de la Narbone, 31062 Toulouse Cedex, France. [6] Nanomaterials and Nanotechnology Research Center, CSIC - Universidad de Oviedo, 33007 Oviedo, Spain. [7] Department of Chemistry, University of Basel, Klingelbergstrasse 82, CH-4056 Basel, Switzerland. [8] Institute for Nanotechnology (INT), Karlsruhe Institute of Technology (KIT), P. O. Box 3640, 76021 Karlsruhe, Germany. [9] Lehn Institute of Functional Materials (LFM), Sun Yat Sen University (SYSU), XinGangXi Rd. 135, 510275 Guangzhou, P. R. China. Correspondence and requests for materials should be addressed to J.F. (email: ferrer@uniovi.es) or to M.C. (email: michel.calame@empa.ch)

Highly-ordered systems combining metal and organic components have recently generated a considerable interest for applications in catalysis, storage and sensing applications. Two-dimensional metal-organic frameworks as electronic systems[1–4] also raise increasing interest thanks to their appealing electric properties, such as high electron mobilities[5] and very large volumetric and areal capacitances[6]. Their one-dimensional counterparts may offer alternative routes for bottom-up nanoscale electronics[7] given the ability of the transition metal centers to change oxidation states and greatly enhance their ability to transport charge[8]. A distinct advantage of these hybrid organic–inorganic systems over all-organic polymers is that they typically do not require dopants to achieve charge transport. Such hybrid materials are typically prepared either by coordination chemistry or by solvo-thermal and hydro-thermal syntheses, providing only limited control over the dimension of the formed oligo- and/or polymers. Moreover, the large influence of the local environment on the chain conformations largely dominates the microscopic dynamics of the growth process, resulting in chain-to-chain variations of the length, tacticity and comonomer incorporation. If such organometallic architectures are to play a role in nanoscale electronics, assembly at the single monomer level is crucial to achieve, as this will allow for an ultimate control over the chain composition and length.

Step-by-step growth of polymers has been first monitored in a protein nanoreactor using spatially separated reactants[9]. Moreover, self-assembled monolayers have been used as a platform to create and control the formation of polymers by alternating precursor molecules[10]. More recently, controlled growth of a single polymer was achieved using magnetic tweezers[11]. Supported by molecular dynamics, the growth dynamics, and in particular the incorporation of new monomers, was studied in detail, with polymer lengths up to several microns.

Oligomerization has also been achieved in molecular devices by the incorporation of metal ions between receptor molecules connected to metal[12] and graphene[13] electrodes or by means of click chemistry[14,15]. However, in these cases the growth process was not monitored in real-time, and a stepwise addition of precursors was required. Moreover, the length of the chains was fixed in advance by the device geometry, and could therefore not be adjusted during the growth process.

Finally, STM measurements on conjugated oligomers chains have been reported in which a chain is lifted up from the substrate using the STM tip[16–19]. In these measurements however, the chains were already preformed on underlying gold the surface. The first signature of in situ dimerization in molecular junctions via covalent bonding was reported using the scanning tunneling microscope break-junction technique[20]. However, no evidence for trimerization or higher-order oligomer formation has been reported.

Here, we demonstrate the controlled formation, unit-by-unit, of single conductive coordination-oligomer chains, up to three units. Key to this achievement is the strong interaction of the isocyano groups with the gold electrodes which leads to a junction formation probability close to unity. This high affinity combined with the strongly dipolar nature of the isocyano anchor allow the molecules in a break junction system to form organometallic chains mediated by the incorporation of gold atoms. These atoms are provided by the electrodes, and do not require the stepwise addition of solutions or precursors. The chains are formed using mechanically controllable break junctions (MCBJ), and the chain formation is monitored in real-time by means of electrical characterization.

## Results

**Electrical characterization.** We perform electrical conductance measurements on the three 1,4-benzenediisocyanide derivatives displayed in Fig. 1a (abbreviated with: BdNC: 1,4-diisocyano-benzene; MBdNC: 1,4-diisocyano-2,5-dimethylbenzene; tBuBdNC: 1,4-diisocyano-2,5-di-*tert*-butylbenzene) using the MCBJ approach in liquid environment[21], of which a schematic layout is depicted in Fig. 1b. In this technique, a nanometer-sized gold wire is broken in the presence of the molecular solution. Upon breaking the gold wire forms atomically sharp electrodes, which the molecules can bridge. After breaking of the gold-molecule junction, the gold contacts are fused again and the process is repeated, allowing to address many different junction configurations. Throughout the entire opening/closing cycle, the current through the junction is monitored. The calibration of the distance is performed as described previously[21]. The measurements are performed at room temperature in 100 μM solution in a 1:4 (v/v) mixture of tetrahydrofuran (THF) and mesitylene (Mes) as solvent. More details concerning the experiments are described in the Methods section.

**Measurements on BdNC.** The central panel of Fig. 1c displays two representative breaking traces recorded on the BdNC molecule, alongside the corresponding conductance-displacement histogram, built from 1445 breaking traces recorded on 6 different samples at a bias voltage of 0.1 V and without any data selection. All breaking traces are aligned in displacement at the rupture point of the last gold–gold contact. As a prominent feature in the conductance-displacement histogram we observe the formation of two plateau-shaped regions of high counts, one at high conductance (plateau 1, abbreviated as P1) and one at low conductance (plateau 2, abbreviated as P2). We also note that a prominent peak is visible at 1 $G_0$, and very few counts down to the P1 plateau, indicative of atomically sharp electrodes. In between the two plateaus, abrupt jumps in conductance are observed, as is apparent from the individual breaking traces. After P2, no distinct region of high counts is present down to the noise floor of our measurement setup, which is around $2 \times 10^{-6}$ $G_0$. The most probable conductance value of each plateau is obtained by fitting the two peaks in the conductance histogram to Gaussian distributions, as illustrated in the right panel of Fig. 1c. The fitted conductance values are $G_{P1} = 10^{-2.1}$ $G_0 = 8 \times 10^{-3}$ $G_0$ and $G_{P2} = 10^{-3.7}$ $G_0 = 2 \times 10^{-4}$ $G_0$ for the P1 and P2 plateau, respectively. Another remarkable observation is that the yield of junction formation is more than 99% (see Supplementary Note 1 for more details). This value is high for molecular junctions, even for liquid environments[22].

BdNC molecules were previously measured in self-assembled monolayers[23], in MCBJ setup at cryogenic temperatures in high vacuum[24] and using the scanning tunneling microscopy break junction (STM-BJ) approach in solution[25–27]. In none of those measurements the formation of two conductance plateaus was reported. However, the P2 plateau in our experiment appears at a conductance lower than the detection limit ($G > 10^{-4}$ $G_0$) of the latter study. More generally speaking, the presence of multiple conductance plateaus or multiple conductance peaks has been observed previously for other molecules and was attributed to multiple molecules in parallel bridging the electrodes[28,29], multiple binding configurations of a single molecule[30–33], switching between internal conductance states[24,34], π–π-stacking[35] or rectification[36] with a different conductance for the forward and reverse current direction.

One way to distinguish between these scenarios is by comparing the length of the breaking traces to the length of the molecule. To extract this plateau length from the opening traces, we developed a plateau detection scheme of which more details can be found in Supplementary Note 2. From this analysis, we obtain an average length for the first and second plateau of 6.4 and 9.0 Å, respectively. Including the retraction of the electrodes

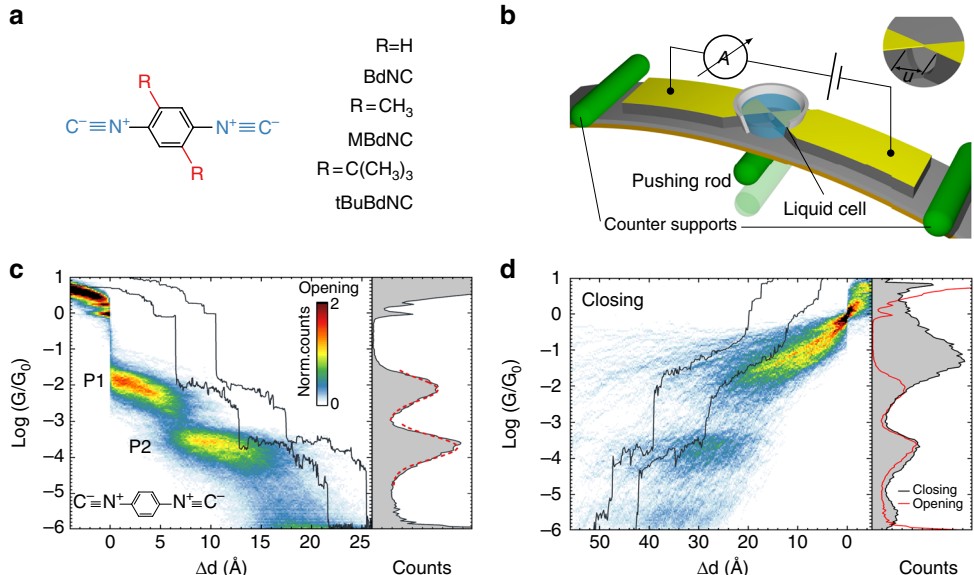

**Fig. 1** Double plateau formation in isocyano compounds. **a** Structural formula of measured compounds. **b** Schematics of MCBJ setup for liquid measurements. **c** Overview of BdNC measurements: 1445 opening conductance traces at 0.1 V bias voltage in 100 μM solution in THF:Mesitylene 1:4 (v/v). Central panel: conductance-displacement histogram and two typical opening traces. The histogram displays the counts per bin on a linear scale with 25 bins per decade of conductance and a bin size of 0.15 Å for displacement. Sample traces are shifted horizontally for clarity. Right panel: conductance histogram with 25 bins per decade. Red dashed curves represent Gaussian fit of the conductance peaks. **d** Overview of closing traces analysis: 1445 traces at 0.1 V bias voltage in 100 μM BdNC solution. Central panel: conductance-displacement histogram and two typical closing traces. Right panel: conductance histogram with 25 bins per decade; red curve: conductance histogram of opening traces shown in (**c**)

of about 5 Å[22], this observation suggests a total electrode separation of about 20 Å at the end of the plateau P2. BdNC has a computed length of 7.8 Å; scenarios involving multiple molecules in series are therefore more feasible than those with a single molecule possessing multiple degrees of freedom binding to the two electrodes (different ground/excited states, different binding geometries, rectification).

**Closing traces analysis**. In order to gain more insight in the nature of the P2 plateau, we also investigated the junction behavior upon closing of the gap between the electrodes. Closing conductance traces have not been widely studied since for most molecules no clear conductance features are observed[37–39]. This is usually attributed to the fact that molecules tend to lay flat on the electrode surface after the breaking of the molecular junction.

Figure 1d displays the closing conductance traces acquired in the same experiment as Fig. 1c, presented in a similar fashion as the their corresponding breaking traces, and aligned at 1 $G_0$. Similar to the breaking, two regions of high counts appear, with a large jump in conductance in between them. Overall, two jumps in conductance are observed: the first one is from the noise level of the setup to an intermediate state with a conductance value close to $G_{P2}$. Here a plateau-like feature is observed, followed by another jump in conductance to a second region of high counts, ranging upwards from the P1 region of the breaking and slanted upwards. For comparison, the breaking histogram is plotted in red on top of the closing histogram in the right panel of Fig. 1d. The P1 peak has a higher conductance and a larger width during the closing. This effect is attributed to the evolution of the electrodes shape through an opening/closing cycle; the electrodes are atomically sharp during opening and more blunt during closing[37]. This is a result of the surface relaxation after the breaking of the molecular junction, and least to more molecules bridging the electrodes during closing. The P2 peak, on the other hand, has a similar position, width, and height for both histograms, suggesting comparable junction geometries are being

probed. Moreover, upward jumps in conductance are visible, both before and after the P2 plateau.

This observation is remarkable, and uncommon for single molecule measurements. The highly reproducible nature of the P2 plateau, combined with its total length close to 20 Å suggests a strong type of interaction between the molecules in series, while π–π-stacking is weak[35,40]. The binding energies for π–π-stacked molecule are typically in the range of 10–100 meV, the same order of magnitude as $k_BT$ at room temperature, making this type of interaction unlikely to yield such reproducible conductance traces. In the following, based on theoretical calculations, we will show that these distinct features, in fact, are indicative of the in situ formation of molecular dimers.

**Theoretical calculations**. For further insights into the dynamics throughout the opening/closing cycles, we have performed Density Functional Theory based Molecular Dynamics (DFT-MD) simulations at room temperature of several molecular junctions formed by BdNC molecules between gold electrodes using the codes SIESTA[41] and GOLLUM[42]. For a detailed description of these calculations, we refer to Supplementary Note 7. In short, the electrodes are pulled apart in steps of 0.1 Å, similar to the method reported previously[43]. The entire system is relaxed by taking 200 DFT-MD steps of 1 fs each after each of those pulling steps. Each of these DFT-MD steps is in turn accomplished as follows: the free energy of the whole system at the desired temperature is calculated using DFT. The force felt by each atom in the system is computed by direct differentiation of the above free energy with respect to the atom displacements. Each atom is moved by solving Newton's classical equations of motion. The whole system is kept at the desired target temperature using a Nose thermostat.

Using this approach we have computed several opening and closing traces of which snapshots are shown in Fig. 2a, c, respectively. From these trajectories, two observations can be made. First, we find that for electrode separations of up to 8 Å after the gold neck is broken, a single molecule bridges the two

 3

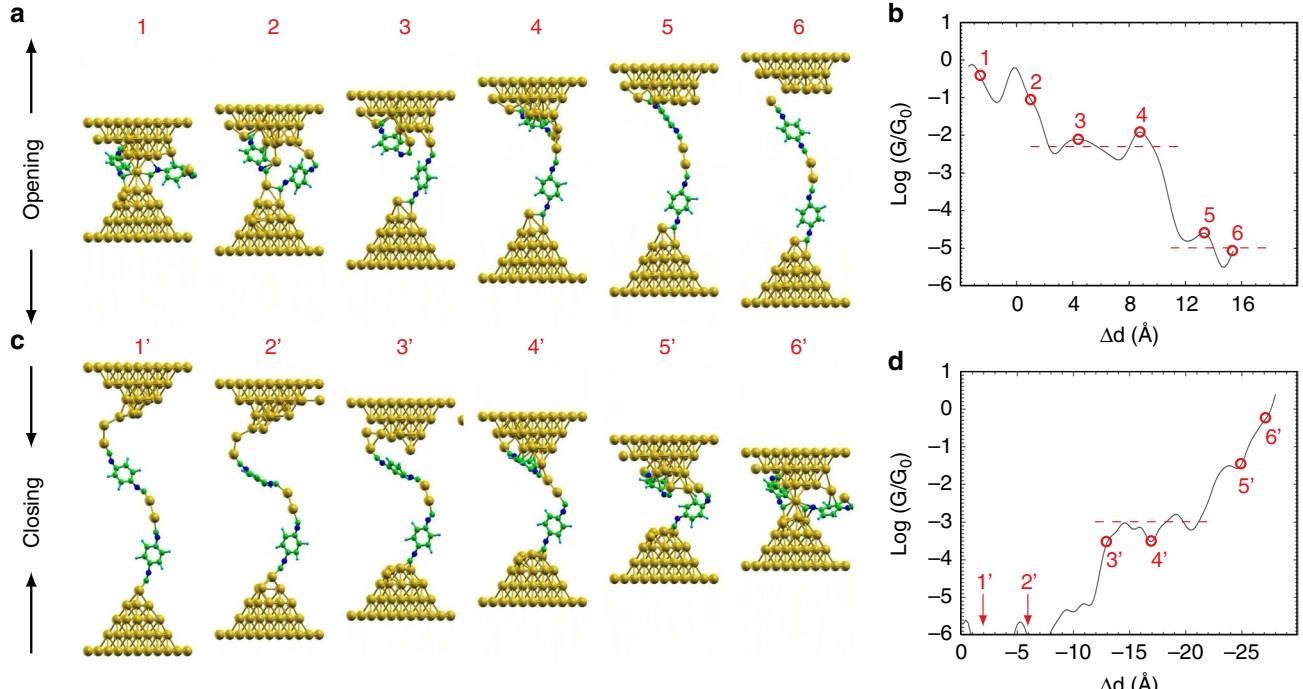

**Fig. 2** Molecular dynamics simulation demonstrating chaining mechanism. **a**, **c** Snapshots of the molecular dynamics simulations for the opening and the closing of the BdNC molecular junction, respectively, demonstrating the chaining effect in presence of extra BdNC molecule. **b**, **d** Reconstructed conductance traces along the corresponding DFT-MD trajectories. Numbered points correspond to the snapshots in (**a**, **c**)

electrodes, while for larger separations the molecules tend to form a dimer chain. In fact, we find that BdNC molecules have a strong tendency to polymerize in situ during the opening of the junction. Moreover, the formation of these chains is always mediated by gold atoms that are pulled out of the electrodes and are being inserted between the BdNC monomers. In contrast, bare isocyano end-groups repel each other due to strong Coulomb interaction of terminal carbon atoms of isocyano-group[44]. Second, the BdNC molecules also tend to remain perpendicular to the gold surface, rather than laying flat on the electrodes.

To rationalize these observations, we performed additional angle-dependent DFT-based total energy calculations (see Supplementary Note 8). These calculations reveal the highly directional nature of the isocyano–Au bond, that possesses a local energy minimum for a contact angle of 90 degrees, in addition to the usual minimum at 0 degrees. We stress that the minimum at 90 degrees is absent for other molecules like 1,4-benzenedithiol or 1,4-dicyanobenzene. This predisposition for large contact angles results in a high packing density of the molecules on the electrode surface with the molecules pointing with one side away from the electrodes. Combined with the very strong binding energy of the isocyano group to the gold, this directionality allows us to rationalize the high yield of molecular junction formation observed in the experimental data.

Large binding energies are also known to promote the extraction of gold atoms from the electrodes. Previous studies on electromechanical properties of molecular junctions have shown that during the breaking of a molecular junction, Au–Au bonds can be broken in the presence of a strong covalent bond between the anchoring group (such as thiols) and gold atoms[45], leading to the pulling of atoms from the electrodes. Another important aspect is that, due to Coulomb repulsion of terminal C atoms, isocyano groups cannot bind directly to other isocyano groups. This particularity therefore makes the formation of an organometallic chain the most plausible scenario and is in line with gold-mediated isocyanide-based organometallic compounds

which have been observed in early crystallography studies[46,47] and have more recently been synthesized and studied in monolayers[48].

Based on the DFT-MD trajectories, we also computed the conductance through the molecular junction as a function of electrode separation (for more detailed information, see Supplementary Notes 9 and 10). The resulting conductance traces are presented in Fig. 2b, d for opening and closing of the electrodes, respectively. For the opening trace, shown in Fig. 2b, two plateaus are visible. The first has a length of about 5 Å and a conductance of about $5 \times 10^{-3}$ $G_0$, while the second one has a conductance of about $10^{-5}$ $G_0$. We note that the difference in conductance between the two plateaus is more than two orders of magnitude. For the closing trace, we find a 10 Å-long plateau with a conductance of about $10^{-3}$ $G_0$.

We have performed 30 such opening traces (see Supplementary Note 11 for more detail), 16 of which ended in the successful formation of a molecular junction. This yield is lower than in the experiments but one should keep in mind that the stretching speed of the MD simulations is much quicker than the experimental one, and that no solvent is present in the calculations, which may reduce the probability of the oligomerization process to occur. Of these molecular junctions, 0 showed the formation of a monomer, while in 5 a dimer was observed. Finally, one trace yielded a monomer and a dimer in parallel, and we count this as a monomer because the two compounds bridge the electrodes in parallel, bringing the total count of monomer traces to 11. A conductance histogram is presented for both scenarios in the Supporting Information. We find that the histogram of the dimers agrees well with the experimental values of P1 and P2 of BdNC at high concentration. Moreover, the conductance histogram for the simulated monomers resembles the experimental one for BdNC at low molecule concentration.

Altogether, the combined results of the DFT-MD simulations and NEGF calculations are in line with experimental data, and point towards a chaining model as most plausible explanation for

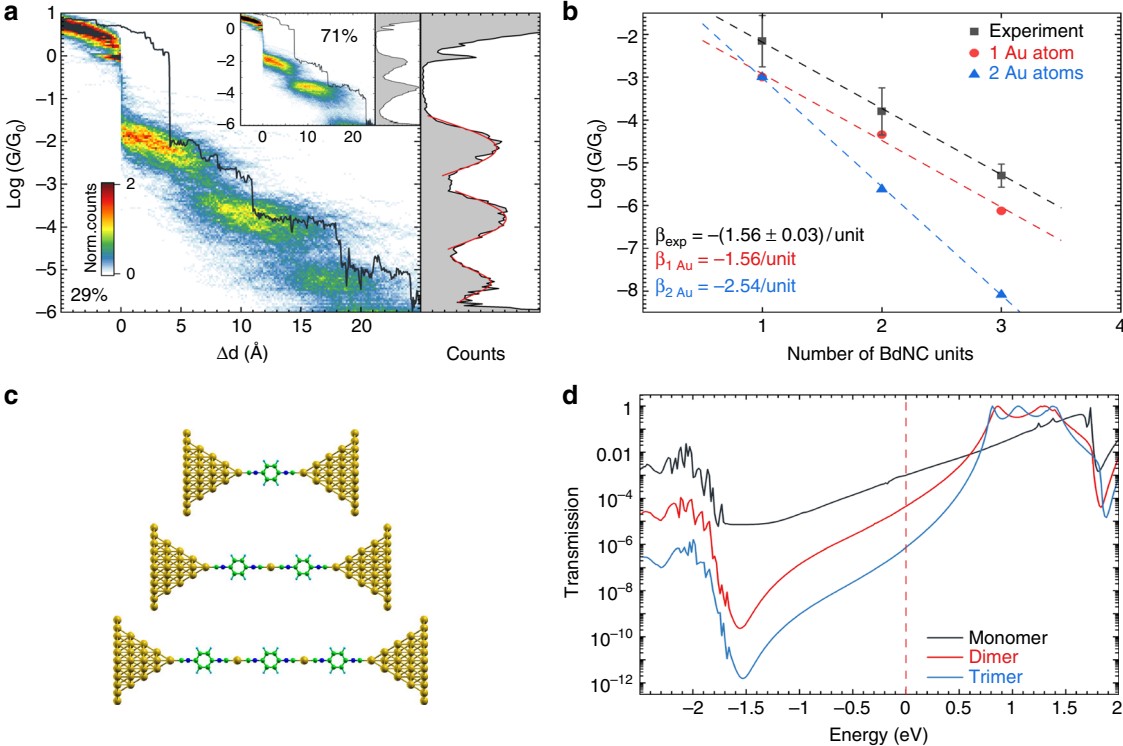

**Fig. 3** Observation of trimer formation. **a** Combined 2D–1D histogram and sample trace for 420 opening conductance traces (29%) for 100 μM BdNC measurements exhibiting third plateau formation. Red curves represent Gaussian fit of the conductance histogram for the plateaus yielding conductance values of $7 \times 10^{-2}$ G$_0$, $1.6 \cdot 10^{-4}$ G$_0$ and $5.0 \times 10^{-6}$ G$_0$ for plateaus P1, P2 and P3, respectively. 2D–1D histogram for remaining 1025 traces (71%) is shown on the inset. **b** Experimental and computed conductance decay versus number of BdNC units. Error bars correspond to the standard deviations of the conductance peaks fit in (**a**). **c** The three junction geometries containing one gold atom as interconnect that have been used to compute the transmission shown in (**d**). **d** Calculated transmission functions for the chains formed by 1–3 BdNC units with one gold atom between units

the formation of the P2 plateau. Indeed, the model provides a justification for the experimental traces being more than twice as long as the length of the molecule by means of incorporating one or two gold atoms during the formation of the organometallic dimer.

**Oligomerization**. Based on the high affinity of isocyanides to form organometallic dimers, we expect the formation of oligomeres with a larger amount of repeating units, such as trimers or tetramers to be plausible. In Fig. 1c, however, no signature for such events are observed, and one may argue that this is either caused by their conductance being below the noise floor of the setup, or their yield being too low to show up in the conductance-displacement histogram. Based on the conductance of plateau P1 and P2, and assuming an exponential dependence of the conductance on the amount of the units, the conductance of the organometallic BdNC trimers is expected to be around $G_{P3} \approx 10^{-5.3}$ G$_0$, and $10^{-6.9}$ G$_0$ for tetramers. While the conductance of the tetramer is outside our experimentally accessible conductance range, the one for the trimer lays well within.

To investigate this scenario and determine the cause for the apparent lack of a third region of high counts, we developed a data filtering scheme to identify plateaus in the conductance region where the trimers are expected be present. For more information about this analysis scheme, we refer to Supplementary Note 3. By applying this selection method on the data shown in Fig. 1, 420 opening conductance traces (29%) were tested positively for the presence of a third plateau. The conductance- and conductance-displacement histograms of the corresponding traces are shown in Fig. 3a, together with a

typical conductance trace. The conductance-displacement histogram of selected traces exhibits three plateau-shaped regions of high counts, arranged in a stair-like fashion, with lengths of 6.4, 8.4 and 3.4 Å, respectively. The three areas of high counts yield distinct peaks in the conductance histogram with conductance values, determined from Gaussian fits, of $7 \times 10^{-2}$ G$_0$, $1.6 \times 10^{-4}$ G$_0$ and $5.0 \times 10^{-6}$ G$_0$, respectively. The conductance decreases exponentially, with a decay constant of 1.56 decade per unit. The remaining 1025 traces that were not selected (71%) are presented as an inset in Fig. 3a. Here, in contrast, only two plateaus are visible, with a clear drop in conductance after the breaking of the second plateau. The ability of gold-capped BdNC molecules to form trimer chains is confirmed by several MD-DFT simulations that we have carried out in presence of simplified gold electrodes.

Figure 3b shows the extracted conductance value as a function of the number of units present in the chain, following an exponential decay of −1.56 decade per unit. To rationalize this trend, we have computed the conductance of BdNC–Au oligomer chains with up to three repeating units using SIESTA and GOLLUM, for the cases where one and two gold atoms interconnect the BdNC units. Figure 3c presents the junction geometries for the monomer, dimer, and trimer chains having one interconnected Au atom, and Fig. 3d displays the respective transmission curves. We refer to the Supplementary Note 13 for the figures corresponding to two inter-connecting gold atoms. The computed transmissions decays are −1.56 and −2.54 for the two cases, respectively. This observation suggests that the monomers may be interconnected by one gold atom, in agreement with the geometry of previously observed complexes[44].

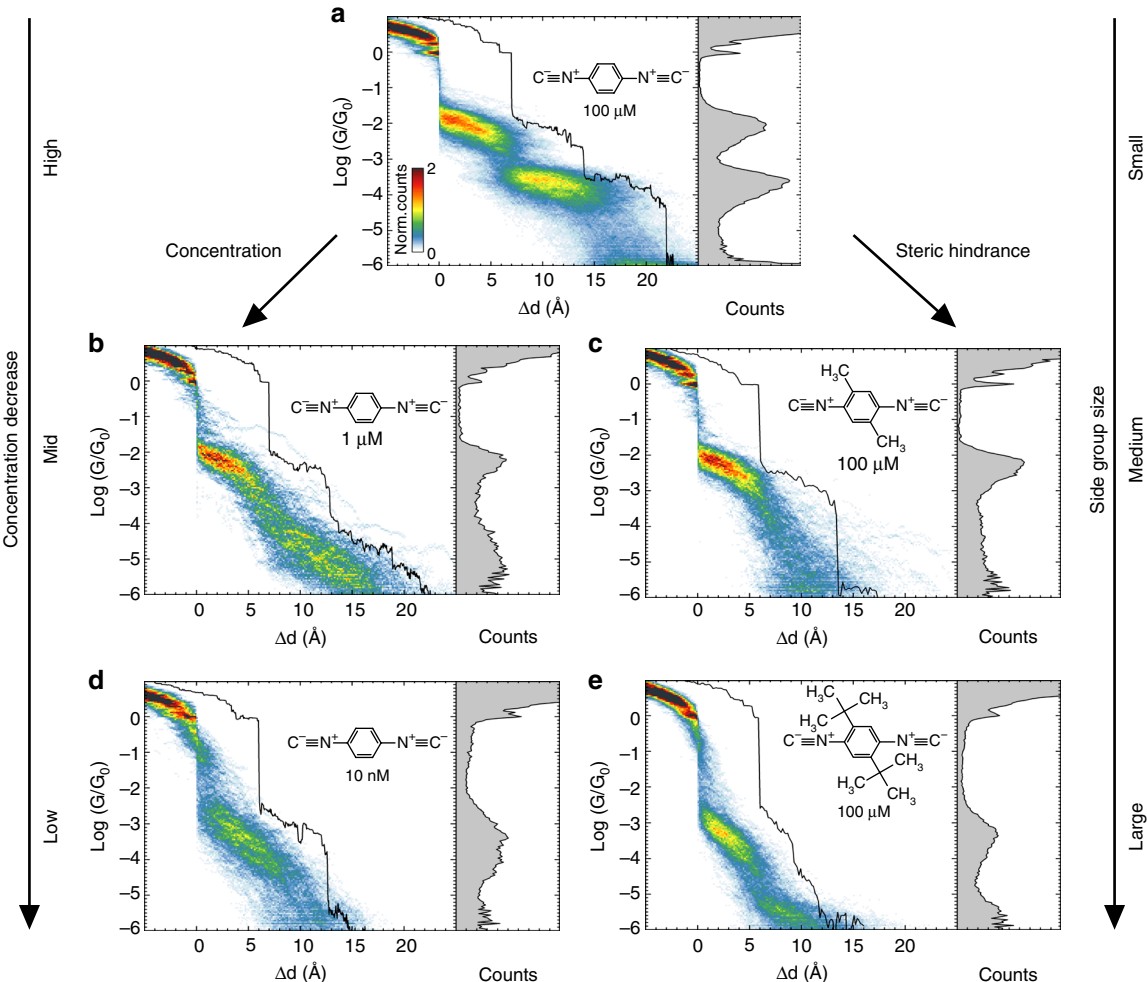

**Fig. 4** Concentration and steric hindrance effect on the chain formation. Combined 2D–1D histograms for (**a**) 100 μM BdNC solution measurement (1445 traces), lower concentration measurements: **b** 1 μM BdNC solution (350 traces), **d** 10 nM BdNC solution (200 traces), and measurements for BdNC derivatives with side groups: **c** 100 μM MBdNC (374 traces) and (**e**) 100 μM tBuBdNC solution (565 traces). Typical opening conductance trace is shown for every measurement

**Surface coverage study**. With the formation of the second and third plateau understood in terms of a chaining effect, the next step is to understand and control the formation of these chains. For chains to form in a repeatable manner, a supply of molecules is required. The dynamics of the chaining process is therefore expected to be determined by the proximity of other molecules, i.e., it should depend on the surface coverage of the electrodes. To investigate the role of the molecular coverage, we employed two approaches. In the first approach, the coverage is influenced by varying the concentration of the molecular solution in which the measurements are performed. In the second one, side groups promoting steric hindrance are added to the molecule.

To investigate the influence of the concentration, we studied the molecular junction formation in solutions with 10 nM and 1 μM concentrations. The first observation is that the signature of the molecular junction formation is distinctively different for each concentration regime, as can be seen in the conductance and conductance-displacement histograms of the corresponding figures (see Fig. 4a, b, d). For a 10 nM concentration, only one slanted conductance plateau is observed, resulting in a peak at $G \approx 10^{-3.1}$ $G_0 = 8 \times 10^{-4}$ $G_0$, with a $\approx 93\%$ junction formation probability. This conductance value is close to the one reported for BdNC molecules in STM-BJ measurements[26,27]. We therefore attribute this plateau to a single- or few-molecule configuration.

For the 1 μM solution, the plateau is less slanted and appears at a conductance value of $10^{-2.3}$ $G_0 = 5 \times 10^{-3}$ $G_0$ with a higher yield of 99%. For a 100 μM solution, Fig. 4a shows the two previously observed plateaus. Finally, to highlight the role of the molecule supply, we also performed measurements in dry conditions (see Supplementary Note 6), in which case very few molecules are expected to be available for the oligomerization process. This rationale is indeed confirmed by the reported measurement which do not show the formation of a dimer but rather a single conductance peak.

The second method to tune the coverage is via steric hindrance. This is achieved by the addition of side groups, and for this purpose, we studied two 2,5-disubstituted derivatives of BdNC: 2,5-dimethyl-1,4-benzenediisocyanide (MBdNC, R=CH$_3$) and 2,5-di-*tert*-butyl-1,4-benzenediisocyanide (tBuBdNC, R=C (CH$_3$)$_3$), with tBuBdNC being the most bulky of the two. The bulky side groups of these molecules reduce the proximity of neighboring molecules at the electrode surface and therefore the availability of an additional molecule for chaining. All measurements were performed in 100 μM solutions of THF:Mes mixture as before.

In the case of MBdNC, we observe one plateau with a conductance value, a peak shape and a plateau length very close to the P1 plateau of the BdNC molecule (Fig. 4c). The yield of

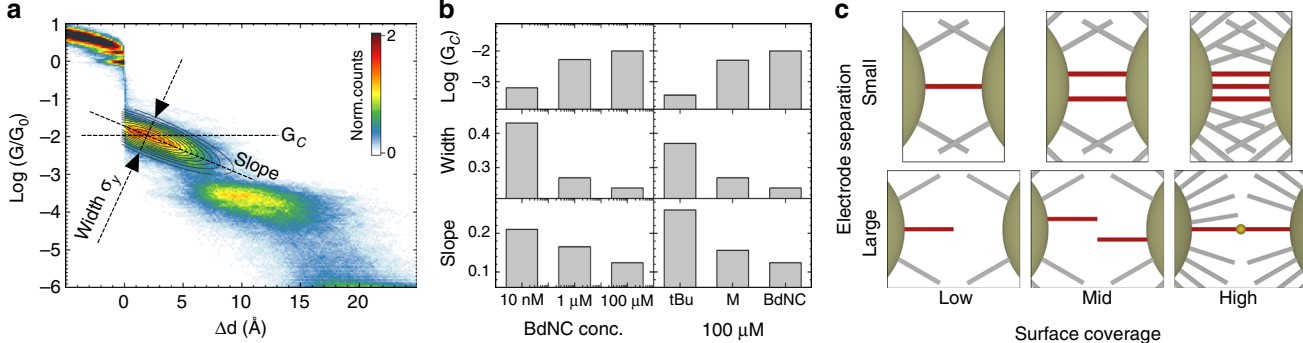

**Fig. 5** Analysis of conductance plateaus. **a** Schematics of plateau shape analysis for 100 μM BdNC measurements. **b** Plateau width, center and slope deduced from the shape analysis (see Supplementary Note 4 for more details). **c** Schematic sketch of the possible scenario's for different surface coverages. Molecules connected to both electrodes are depicted in red, while the others are colored gray

junction formation is almost 100%. For the tBuBdNC molecule, only one short and slanted conductance plateau ($G \approx 5 \times 10^{-4}$ $G_0$) is observed with a lower yield of 91.3% (Fig. 4e). Qualitatively, the signatures of the molecular junction formation for MBdNC and tBuBdNC resemble the ones obtained for the 1 μM and 10 nM solutions of BdNC, respectively.

Overall, both the concentration and the side group study demonstrate that the molecular coverage of the electrode surface is of paramount importance for the formation of the P2 and P3 plateau, i.e., chaining is only possible when additional molecules are sufficiently closeby at the moment the molecular bridge breaks.

**Plateau analysis**. Another interesting observation from Fig. 4 is that the molecular supply not only affects the polymerization process, but also the shape of the P1 conductance plateau. The influence of the coverage on the plateau shape may shed some light on the dynamics, and for this purpose we performed an additional shape analysis of plateau P1 for the various molecules and concentrations in order to quantify the plateau width, slope, and center (see Supplementary Note 4 for more details). Determining the plateau properties was done by fitting the areas of high counts in the conductance-displacement histograms with two-dimensional Gaussians, as illustrated in Fig. 5a. The extracted values are shown in Fig. 5b. The plot shows that both the plateau width and slope monotonously decrease with increasing concentration and decreasing side group size, while the center, on the other hand, moves towards higher conductance values.

The apparent similarity of the plateaus is confirmed by a calculated correlation of the plateau properties for the two different approaches for tuning the surface coverage (0.997, 0.964, and 0.9997 for the width, slope, center, respectively). This correlation motivates us to search for a single mechanism to explain the data from both approaches. The observed changes in junction formation with surface coverage shown in Fig. 5b can be rationalized by considering the role of inter-molecular interactions. Let us first focus on the plateau width. The reduction in plateau width is a result of a decrease in conductance fluctuations within each conductance trace (see Supplementary Note 5), and is indicative for the formation of more stable molecular junction. This most likely results from a reduction of mechanical degrees of freedom due to the formation of molecular stacks composed of a few molecules bridging the electrodes in parallel. One can argue along similar lines when considering the downward orientation of the plateaus. The pronounced slope may be due to sliding of a single molecule on the electrodes during opening of the junction, the probability of which is being reduced when decreasing the number of mechanical degrees of freedom when forming

molecular stacks. This hypothesis concerning the formation of stacks is corroborated by the upwards shift in the center of the plateaus with increasing coverage, a signature of parallel conduction though an increased number of molecules. These scenarios are schematically depicted in Fig. 5c. For low and medium coverage, only single, or few molecules bridge the gap, while for high densities molecular stacks are formed for small electrode separations, followed by dimers and trimers upon increase of the electrode spacing.

## Discussion

It is important to keep in mind that the high molecular coverage is a result of the directionality of the isocyano–Au bond, as it allows for densely packed arrays of molecules. This density is determined by the concentration of molecules in solution. For high coverages, this results in the formation of dimers and trimers, suggesting the experimental conditions play a crucial role in the polymerization process. This strict criterium may explain why previous studies have not observed this chaining phenomenon, as measurements were either performed in dry conditions, or at lower concentrations.

Another point which should be noted is that the formation probability of trimers of 29% may be increased by performing the measurements in a molecular solution of higher concentration. However, this approach has as drawback that for high surface coverages, the junction tends to have difficulties closing to full contact, thereby severely limiting the number of opening traces which can be acquired on a junction.

In the present study, the metallic atoms to form the coordination-oligomer chains were provided by the electrodes by means of extraction during the opening of the junction. This conclusion is supported by the lack of the P2 plateau at low concentrations, which would be present in case the in-solution formation of Au-mediated dimers would play a role. Alternatively, one could also engineer the junctions such that the metallic atoms are supplied by ions present in solution. This approach would allow for more flexibility in terms of available metal centers, and for instance allow to incorporate magnetic atoms for spintronics studies in metalorganic spin-chains, while keeping the convenience of gold electrodes for contacting the chains. Finally, our approach for high surface coverage may be more widely exploited for the realization of mechanically stabilizing molecular junctions, both with the prospect of reducing electronic variability, as well as increasing the robustness in terms of electrode separation.

In summary, we demonstrate the controlled formation of organometallic oligomer chains in solution up to three repeating units. This is achieved by utilizing isocyano anchoring groups,

which combine a highly directional binding with a strong binding energy. The anisotropic binding allows for high surface coverages and yield of junction formation, while the strong binding leads to the extraction of gold atoms from the electrodes, thereby acting as supply of metal ions. We show that the formation of such coordination-oligomer chains can be affected by varying the molecular supply, either by reducing the concentration of molecules, or by the addition of steric side-groups. The influence on the molecular supply provides insights in the formation process of the molecular junctions and points towards the formation of molecular stacks of several molecules in parallel. Our findings are supported by DFT + NEGF calculations, demonstrating both the crucial role of the anisotropic binding, as well as the importance of the metal ions in the chain. An interesting side-effect of the high packing density is the reduction of fluctuations in junction conductivity

## Methods

**Conductance histograms**. The experiments are performed using the mechanically-controllable break-junction (MCBJ) approach in solution at room temperature. The MCBJ samples were fabricated as described previously[35]. In short, a suspended gold bridge of about 500 to 700 nm was fabricated with a 60 nm wide constriction in the center, for a film thickness of 60 nm. To reduce parasitic currents in liquid, an additional layer of 3 μm thickness of photodefinable polyimide was deposited to cover the electrodes, while keeping the gold bridge itself exposed. All molecular solutions were freshly prepared for each measurement using the mixture of THF and Mes (ratio 1:4 v/v) as a solvent. For each sample, the gold bridge was initially broken in pure solvent and 50 opening-closing cycles were recorded. After that, the liquid cell was filled with the target molecular solution. For every cycle, the junction was opened until the conductance of $10^{-6}$ $G_0$ (noise level of the setup) was reached, followed by a subsequent closing of the gold contact (~10 $G_0$). The speed of the pushing rod was set to 31.2 μm/s. The measurements were performed with a sampling rate of 500 Hz and a bias voltage of 0.1 V. The obtained opening and closing traces were analyzed without any data selection.

Opening traces were aligned at the breaking of the last atomic gold contact. Conductance histogram and conductance-displacement histogram were build by bining the opening traces both in distance and in conductance on log scale, using 67 bins per nanometer and 25 bins per decade of conductance. The number of counts then was normalized by the number of traces and sizes of the bins. Due to discreteness of the analogue to digital converter and log-conversion, a few artifacts (spikes in conductance histogram and stripes in conductance-displacement histogram) occur for the conductances below $10^{-5.5}$ $G_0$. The closing conductance traces are treated as reversed opening traces and analyzed in the same way. To build conductance-displacement histogram for the closing traces, the traces were aligned at the point with the conductance of $G_0$.

## Data and code availability

MCBJ measurements were performed with custom-made Labview routines and analysed in Matlab. All relevant data sets and codes are available from the authors upon reasonable request.

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

## Acknowledgements

This work was partially supported the EC FP7-ITN MOLESCO grant (No. 606728). M.P. acknowledges funding by the EMPAPOSTDOCS-II programme which is financed by the European Unions Horizon 2020 research and innovation programme under the Marie Skłodowska-Curie grant agreement number 754364. J.O. and M.C. gratefully acknowledge financial support from the Swiss Nanoscience Institute. Generous support by the Swiss National Science Foundation (SNF grant No. 00020-159730) is gratefully acknowledged by M.G. and M.M. The authors thank Fabian Oppliger and Simon Josephy for assistance with the MCBJ measurements in dry conditions.

## Author contributions

A.V., J.O., and J.B. conducted the liquid-MCBJ measurements. A.V. performed the data analysis. M.G. and M.M. performed the molecular synthesis. R.F., V.G.S, J.F., and M.P. performed the DFT calculations. M.M., J.F., and M.C. designed and supervised the study. A.V., M.P., J.O., J.F., and M.C. wrote the manuscript. All authors participated in the discussion of the data and commented on the manuscript.

## Additional information

**Competing interests:** The authors declare no competing interests.

