## [Peer Review File · Nature Communications]

Reviewers' comments:

Reviewer #1 (Remarks to the Author):

This work done by A. Vladyka et al. demonstrated the bottom-up in-situ formation of two or three organometallic oligomer chains at the single-molecule level by using a mechanically controllable break junction technique. They claimed that, in conjunction with the theoretical calculations, concentration and side chain dependent experiments were performed to identify the oligomerization process and thus the observed different conductance plateau. I feel interested in reading this study, however, the flaws and the lack of novelty existing in the manuscript hamper me to recommend publication in the present presentation. I would like to suggest resubmission after major revisions as follows to another more specific journals, such as Small or Nanoscale.

- 1, The most issue of this work is its novelty since many studies have been carried out using BdNC molecules, either the conductance and the covalent nature between them and the electrodes, as well as the the oligomerization process, as the authors claimed in their article.
- 2, Another important issue existing in this work is the lack of the evidence (both experimental and theoretical) to demonstrate how and how many gold atoms bind to individual BdNC molecules during the oligomerization process.
- 3, The dependence of conductance on the molecular length should be analysed and gave in the manuscript. And compare their data with those reported in the previous work.
- 4, In the introduction, the authors claimed the universal phenomena of multiple conductance plateaus in single molecular junctions. I agree with this comment, but they failed to cite several important literature in the fields (as well as those related to the assembly process). This point is also important to inform readers the recent advances.

Reviewer #2 (Remarks to the Author):

The manuscript by Vladyka et al describes a combined experimental and theoretical work to understand the conductance properties of mechanically controllable break junctions (MCBJ) operated in liquid environments with 1,4-benzenediisocyanide derivatives. Conductance-displacement histograms built from the collected data are presented for a class of three related molecules (BdNC, MBdNC, and tBuBdNC at 100uM) and for three different BdNC concentrations (100uM, 1uM, and 10nM).

The main feature is that only the histograms for the high-concentration BdNC present two (or tree) plateaus. This is interpreted as the in-situ formation of organometallic chains between the electrodes containing two (or three) BdNC units bonded together via gold atoms. It is argued that this chaining process is absent for the other two species and for the lower concentrations due to steric hindrance and likelihood of encountering additional units in the formation process.

The existence of multiple plateaus in the histograms is often interpreted as the signature of the number of molecules bridging the electrodes or different bonding configurations to the electrodes. Here, however, the relatively long distances (compared to the molecular unit length) for the second and third plateaus make such scenarios unlikely.

The experimental results are supported by atomistic calculations that simulate the opening and closing processes via room-temperature molecular dynamics. These simulations show that the proposed chaining process also occurs in model situations. Furthermore, the exponential decay of the

transmission with number of molecular units in the chain is qualitatively in agreement with DFT-NEGF calculations.

The paper reads well, the research is well executed, and the proposed scenario of formation of organometallic chains is credible. As mentioned in the manuscript, in-situ dimerization via covalent bonding was attributed to features reported in Ref. 11. But here the MCBJ approach is clearly taken farther by identifying also trimers and by studying factors that suppress the chaining process. On the other hand, somewhat related lift-off experiments with STM on conjugated oligomers have already shown oscillations in the conductance traces corresponding to one unit after the other being detached from the surface.

I have some questions/comments:

1) The authors argue that the wires form during the pulling process, but could it be that the organometallic wires are actually formed on the electrodes before the pulling sequence starts? Or put differently, what is the evidence that the molecular chains break up during the closing phase?

2) As mentioned above, lift-off experiments with STM on conjugated oligomers were reported in, e.g., Lafferenz et al, *Science* 323, 1193 (2007), Kawai et al, *PNAS* 111, 3968 (2014), Reecht et al, *JPLCL* 6, 2987 (2015). I suggest that the authors to make link to this research.

3) On page 9 it is mentioned that "in some cases, contact between the molecule and the electrodes is made via the phenyl ring, resulting in higher conductance values". To give the reader a more precise idea of this bonding scenario I suggest to show the geometry in the SI. Why is there no experimental signature of this?

4) A similar MD scheme for single-molecule MCBJ was presented in Paulsson et al, *Nano Lett.* 9, 117 (2009). Are there any essential differences in the methods?

5) The authors mention that "a sizeable number" of MD-MCBJ cycles were carried out for BdNC. To average out effects of thermal fluctuations and to improve sampling of the most characteristic configurations, I suggest to construct a conductance histogram from all simulations.

6) Did any of the other molecules studied with MD ever form wires of two or more units?

7) How precisely is the experimental electrode displacement known? How is it calibrated?

8) Could the likelihood of forming chains perhaps also be controlled by the pulling speed (to give more or less time for the molecular units to diffuse to the junction region)?

Reviewer #3 (Remarks to the Author):

In this manuscript, the authors studied in-situ formation of 1D organometallic chains, under the experimental condition of mechanically controlled break junction experiments. The paper reports new phenomena in the field of molecular junctions, and is well organized and well written. The conclusions are well supported by the data. I recommend publication after the following three questions are properly addressed:

(1) It looks to me that two different computing programs, [Redacted], are used in transport

calculations at different places of the paper. I found this practice inconsistent and confusing, and the paper does not justify the necessity of using two different programs for different systems studied in this paper. I suggest that either the authors perform all calculations with a single computing program, or add sufficient explanations and justifications regarding the choice of computing program for specific systems studied in this work;

(2) In the introduction, the authors discussed the need for “an ultimate control over chain composition and length”, and claimed “controlled formation” of the chain. I am worrying that these statements exaggerate what this work has accomplished. For example, it looks to me that the authors could not selectively form a dimer or a trimer, and any formation of a trimer must follow the formation of a dimer first. I suggest the authors adjust the wording in the introduction to correctly reflect what has been actually accomplished in this work;

(3) The authors studied concentrations of 100 μM , 1 μM , and 10 nm . How about concentrations larger than 100 μM ? Can the authors form even longer chains such as tetramers? It would be great to know the limit of this approach.

Reviewer #1 (Remarks to the Author):

This work done by A. Vladyka et al. demonstrated the bottom-up in-situ formation of two or three organometallic oligomer chains at the single-molecule level by using a mechanically controllable break junction technique. They claimed that, in conjunction with the theoretical calculations, concentration and side chain dependent experiments were performed to identify the oligomerization process and thus the observed different conductance plateau. I feel interested in reading this study, however, the flaws and the lack of novelty existing in the manuscript hamper me to recommend publication in the present presentation. I would like to suggest resubmission after major revisions as follows to another more specific journals, such as Small or Nanoscale.

1, The most issue of this work is its novelty since many studies have been carried out using BdNC molecules, either the conductance and the covalent nature between them and the electrodes, as well as the the oligomerization process, as the authors claimed in their article.

We agree with the referee that several studies have been performed on BdNC (see for instance references 40 and 41 of the manuscript). However, none of those studies reported about the *in-situ* oligomerization of molecules evidenced in transport measurements. To the best of our knowledge, this was never directly observed. Therefore, we do not agree with the referee concerning the lack of novelty. We also note that Reviewer #3 states "The paper reports new phenomena in the field of molecular junctions". Should we have missed a relevant publication demonstrating this oligomerization process as demonstrated here, we would be grateful if the reviewer could provide us with a reference.

An important difference with published studies is the fact that we perform our measurements in liquid and at very high molecular concentration. As described in the manuscript, the concentration is key in achieving oligomerization. To verify this claim, we have performed additional measurements in dry conditions, and find that in this case no dimers are formed. These measurements have been added to the Supporting Information.

2, Another important issue existing in this work is the lack of the evidence (both experimental and theoretical) to demonstrate how and how many gold atoms bind to individual BdNC molecules during the oligomerization process.

The incorporation of a gold atom in the chains occurs by extraction of the atoms from the electrodes. Our DFT+MD calculations clearly demonstrate this. Extraction of gold atoms from the electrodes is not uncommon for other anchoring groups and has long been known for thiols (see for instance Xue et al. Nat. Comm. 5, 2014). The isocyano-Au interaction is even stronger than that of thiols, corroborating our findings.

Concerning the number of gold atoms in the chain, we know that from a chemical point of view at least one gold atom should be present as isocyano groups are strongly polar and repel each other. Therefore,

they cannot directly bind one to another. The presence of a gold atom is confirmed by the opening cycles we computed using DFT+MD in which we observe that either one or two gold atoms can be incorporated into the dimer. . Furthermore, we carried out several Molecular Dynamics simulations where we tried to bind two BdNC molecules that did not possess gold-capping atoms. We found in all cases that the molecules did not bind and furthermore repelled each other. We are therefore confident that at least one gold atom is required for the formation of dimers/trimers.

More specifically, we performed additional analysis, and believe that only one atoms is present in the dimer. Experimentally, we observe for BdNC at high concentration that the length of the second plateau is slightly longer than the length of the molecule (9.0Å, versus 7.8Å for the molecule length), which is indicative of one gold atom (approx. 2.5Å) being incorporated into the chain. Moreover, from a comparison with the DFT+MD calculations, the experimental decay constant per unit is closer to the one computed for one gold atom than for two. We have verified this comparison using two different codes: [Redacted] , and have found that they agree with each other and both decay constant consistently point towards one gold atom. Finally, isocyanobenzene molecules have been reported to form linear cations mediated by a single Au(I) ion, (see e.g. the x-ray structure in: W. Schneider, A. Sladek, A. Bauer, K. Angermaier, H. Schmidbauer, Zeitschrift für Naturforschung, B: Chemical Sciences (1997), 52(1), 53-56).

[Redacted]

Figure R1. X-ray structure of the dimer consisting of two isocyanobenzenes interlinked by an Au(I) ion. The figure displays the ORTEP plot with 50% probability ellipsoids and is taken from the article mentioned above (W. Schneider et al., Z. Naturforsch. 52b, (1997), 53-56).

3, The dependence of conductance on the molecular length should be analysed and gave in the manuscript. And compare their data with those reported in the previous work.

We thank the referee for this suggestion. A plot of the conductance versus molecular length has been added to Fig.3 for the monomer, dimer and trimer, together with the calculated conductance values. In the manuscript, a comparison with the reported literature values of the monomer conductance is provided. As mentioned previously, we are not aware of any reports in literature on the *in-situ* formation of dimers/trimers with BdNC.

4, In the introduction, the authors claimed the universal phenomena of multiple conductance plateaus in single molecular junctions. I agree with this comment, but they failed to cite several important literature in the fields (as well as those related to the assembly process). This point is also important to inform readers the recent advances.

Multiple plateaus have been observed in break-junction measurements, which have been explained by various mechanisms (see references 17-27). It is true that many other papers have been reporting similar scenarios, but we believe that the selected papers cover all possible scenarios discussed to date. Should we have missed another mechanism described in the literature, we will be happy to include it in the list as well.

Reviewer #2 (Remarks to the Author):

The manuscript by Vladyka et al describes a combined experimental and theoretical work to understand the conductance properties of mechanically controllable break junctions (MCBJ) operated in liquid environments with 1,4-benzenediisocyanide derivatives. Conductance-displacement histograms built from the collected data are presented for a class of three related molecules (BdNC, MBdNC, and tBuBdNC at 100uM) and for three different BdNC concentrations (100uM, 1uM, and 10nM).

The main feature is that only the histograms for the high-concentration BdNC present two (or tree) plateaus. This is interpreted as the in-situ formation of organometallic chains between the electrodes containing two (or three) BdNC units bonded together via gold atoms. It is argued that this chaining process is absent for the other two species and for the lower concentrations due to steric hindrance and likelihood of encountering additional units in the formation process.

The existence of multiple plateaus in the histograms is often interpreted as the signature of the number of molecules bridging the electrodes or different bonding configurations to the electrodes. Here, however, the relatively long distances (compared to the molecular unit length) for the second and third plateaus make such scenarios unlikely.

The experimental results are supported by atomistic calculations that simulate the opening and closing processes via room-temperature molecular dynamics. These simulations show that the proposed chaining process also occurs in model situations. Furthermore, the exponential decay of the transmission with number of molecular units in the chain is qualitatively in agreement with DFT-NEGF calculations.

The paper reads well, the research is well executed, and the proposed scenario of formation of organometallic chains is credible. As mentioned in the manuscript, in-situ dimerization via covalent bonding was attributed to features reported in Ref. 11. But here the MCBJ approach is clearly taken

farther by identifying also trimers and by studying factors that suppress the chaining process. On the other hand, somewhat related lift-off experiments with STM on conjugated oligomers have already shown oscillations in the conductance traces corresponding to one unit after the other being detached from the surface.

We would like to thank the referee for the encouraging comments and are happy to hear that the research is well executed.

I have some questions/comments:

1) The authors argue that the wires form during the pulling process, but could it be that the organometallic wires are actually formed on the electrodes before the pulling sequence starts? Or put differently, what is the evidence that the molecular chains break up during the closing phase?

Direct experimental proof of the breaking of the dimer is very challenging to obtain both in the opening as in the closing. Based on our DFT+MD calculations, we can say that the oligomer is preserved during the opening process, and that the molecular junction is broken in all cases at the contacts, not at the gold atom(s) within the chain. However, the same calculations also show that the dimer can be broken when fusing the electrodes. The Figures R2 and R3 below display full closing/opening cycles that illustrate both scenarios. In Fig. R2 the dimer is broken upon fusing and does not reform upon successive opening, while in Fig. R3 it does reform.

Figure R3. A closing/opening cycle where the dimer is destroyed but reforms afterwards.

Figure R3. A closing/opening cycle where the dimer is destroyed and reforms again.

2) As mentioned above, lift-off experiments with STM on conjugated oligomers were reported in, e.g., Lafferenz et al, Science 323, 1193 (2007), Kawai et al, PNAS 111, 3968 (2014), Reecht et al, JPCL 6, 2987 (2015). I suggest that the authors to make link to this research.

We would like to thank the referee for pointing this out. Lift-off experiments have indeed been reported on conjugated oligomers and we overlooked citing this approach. We have therefore added several references to the manuscript and added a sentence in the introduction to make the link with this research field.

3) On page 9 it is mentioned that "in some cases, contact between the molecule and the electrodes is made via the phenyl ring, resulting in higher conductance values". To give the reader a more precise idea of this bonding scenario I suggest to show the geometry in the SI. Why is there no experimental signature of this?

The formulation in the text may have been confusing. The scenario in which the phenyl ring binds to the gold are observed in the closing cycle calculations. Fig. R4 illustrates a few relevant stages of an MCBJ closing cycle where a phenyl ring is contacting the electrode, until a gold neck is formed. To avoid confusion, we have removed the concerning comment in the text.

Figure R4. Three stages of a closing cycle for a BdNC molecule, where the phenyl ring is contacted first instead of the capping gold atoms.

4) A similar MD scheme for single-molecule MCBJ was presented in Paulsson et al, Nano Lett. 9, 117 (2009). Are there any essential differences in the methods?

We followed the same procedure described in Paulsson's article. However, we found that Paulsson's 60 MD steps were not nearly enough in our case because the pyramids became completely destroyed and one or several gold filaments were created in every case. We found that we needed at least 200 MD steps to retain enough adiabaticity so that the pyramids retained its identity and gold filaments were suppressed. This is the protocol that we adopted in all the MD calculations discussed in the manuscript. We have clarified this point in the main text when describing the theoretical method.

5) The authors mention that "a sizeable number" of MD-MCBJ cycles were carried out for BdNC. To average out effects of thermal fluctuations and to improve sampling of the most characteristic configurations, I suggest to construct a conductance histogram from all simulations.

In total, we performed 30 opening cycles, of which 16 made a molecular junction (53 % yield). We found that 11 of them consisted of monomers and 5 of dimers. We have added in the SI the conductance histograms of these two binding geometries. We find that the histogram of the dimers agrees well with the experimental values of P1 and P2 of BdNC at high concentration. Moreover, the conductance histogram for the simulated monomers resembles the experimental one for BdNC at low molecule concentration. These statistics are now discussed in the main text.

6) Did any of the other molecules studied with MD ever form wires of two or more units?

From a statistical point of view, to be able to exclude dimer formation would require a prohibitively large number of opening traces. As mentioned in the reply to referee 1, out of 30 such opening traces with BdNC, 5 show the formation of a dimer. As we expect a lower dimer formation yield for tBu-BdNC due to steric hindrance effects (more on this below), a significantly larger amount of opening traces would be needed

As an alternative approach, we performed DFT+MD calculations to investigate the interaction between 2 tBu-BdNC, molecules, of which two examples are shown in Figure R5. The plot displays two relaxed

junction geometries, which the steric effects are clearly visible. A significantly lower junctions formation yield is therefore expected.

Figure R5.MD DFT of two junction geometries, each containing 2 tBu-BdNC molecules.

7) How precisely is the experimental electrode displacement known? How is it calibrated?

The electrode displacement has been determined for this electrode geometry by analyzing the tunneling decay in solvent. The full procedure can be found in reference 16 of the manuscript. We have added a sentence in the text to clarify this point.

8) Could the likelihood of forming chains perhaps also be controlled by the pulling speed (to give more or less time for the molecular units to diffuse to the junction region)?

This is an interesting question, which we have not investigated in detail. The actual measurements were all performed at the maximum electrode speed our stepper motor allows us to go. We believe that for very slow measurements, additional reorganizations at the electrode level may play a role. To investigate the dynamics of dimer formation we therefore deemed our level of control to be better by varying the molecular concentration and the addition of bulky side groups.

Reviewer #3 (Remarks to the Author):

In this manuscript, the authors studied in-situ formation of 1D organometallic chains, under the experimental condition of mechanically controlled break junction experiments. The paper reports new phenomena in the field of molecular junctions, and is well organized and well written. The conclusions are well supported by the data. I recommend publication after the following three questions are properly addressed:

We would like to thank the referee for the encouraging comments and are happy to read that the reviewer agrees that the reported phenomena are new and that the conclusions are well supported by the data.

(1) It looks to me that two different computing programs, [Redacted], are used in transport calculations at different places of the paper. I found this practice inconsistent and confusing, and the paper does not justify the necessity of using two different programs for different systems studied in this paper. I suggest that either the authors perform all calculations with a single computing program, or add sufficient explanations and justifications regarding the choice of computing program for specific systems studied in this work;

We agree with the referee that the use of two different codes may be confusing. We therefore repeated the calculations performed in [Redacted], with no difference in the conclusions to be drawn from them. The paper now only contains calculations using Siesta+GOLLUM, which predict decay constants for 1 Au and 2 Au atoms to be -1.56 and -2.54 decade per unit, respectively.

We note that this agreement between [Redacted] provide further robustness to the theoretical calculations that we have carried out, as well as to the agreement between theory and experiments. The manuscript and supporting information have been adapted where needed.

(2) In the introduction, the authors discussed the need for “an ultimate control over chain composition and length”, and claimed “controlled formation” of the chain. I am worrying that these statements exaggerate what this work has accomplished. For example, it looks to me that the authors could not selectively form a dimer or a trimer, and any formation of a trimer must follow the formation of a dimer first. I suggest the authors adjust the wording in the introduction to correctly reflect what has been actually accomplished in this work;

We indeed discuss the need for “an ultimate control over chain composition and length”. However, we do not claim that we achieved this and in particular not the chain composition, as we do not know whether 1 or 2 gold atoms are incorporated in the chain. What we do claim is the controlled formation of chains. We show in the manuscript that with our approach we can form monomers, dimers and trimers. It is true that we first need to trap a monomer before forming a dimer and a dimer before a trimer, but we do control the formation of the dimer/trimer, as we can stop the electrodes at any moment along the chain formation process

(3) The authors studied concentrations of 100 μM , 1 μM , and 10nm. How about concentrations larger than 100 μM ? Can the authors form even longer chains such as tetramers? It would be great to know the limit of this approach.

At concentrations larger than 100 μM , we observe that after only few cycles the junctions cannot be closed anymore. This is due to an overcrowding or clogging effect in the gap, which prevents the fusing of the electrodes. As a closing of the electrodes up to a conductance of a few G_0 is necessary to ensure a

proper resetting of the junctions, we limited the maximum concentration to 100mM. The presence of tetramers is an interesting question. The predicted conductance based on the experimental data would be around $3e-7$ G0. Unfortunately, this is below the noise floor of our liquid-based measurement setup. It would be interesting to investigate such a system using for instance conducting AFM measurements, in which the force can be used to monitor the oligomer formation, even after its conductance decreases below the detection limit. .

REVIEWERS' COMMENTS:

Reviewer #1 (Remarks to the Author):

I have a chance to look through the revised manuscript with interest. I thank the authors to make changes with efforts. After carefully reading the manuscript, I am kind of disappointed about their arguments and knowledge. Listed below are my scientific and irrefutable criticisms:

One major concern is still the lack of novelty existing in the manuscript. The authors asked me to provide the relevant publications demonstrating this self-assembly/polymerization process. Here are two examples: *Angew. Chem. Int. Ed.* 2009, 48, 5178–5181, which demonstrated the concept by synthesizing fluorene oligomers of different lengths within the gaps and studying their transport properties; Similarly in the same journal, *Angew. Chem. Int. Ed.* 2007, 46, 3892–3895, which detailed a method to prepare multicomponent electrical circuits through the nanoscale placement of reactive groups through coordination or condensation reactions. Obviously, there are more reports related to this topic. Indeed, the concept of oligomerizing molecules evidenced in transport measurements is not new. I strongly and kindly recommend the authors to read and learn the latest advances in the field.

In addition to the novelty issue, another significant issue is the unconvincing evidence for the in-situ formation of molecular junctions as described by the authors. The authors cited a reference showing the structure with one Au as the bridge. I agree with this structure. However, they failed to provide any evidence to demonstrate two Au atoms between each BdNC unit. This is suspicious. Furthermore, this experiments seems that the Au atoms on the tip are mobile, which leaves me with an impression that the devices are unstable. The device instability is fatal and will not be useful for further applications.

Overall, the flaw and novelty issues existing in the manuscript still hamper me to recommend publication in the prestigious journal of *Nature Communications*. I hope that my comments help the authors to revise their manuscript for resubmission to the more specific journal.

Reviewer #2 (Remarks to the Author):

The authors' responses to the reviewer comments and the corresponding manuscript revisions are all very satisfactory. I recommend publication in the current form.

Reviewer #3 (Remarks to the Author):

In a revised manuscript, the authors have addressed all my concerns and questions from a previous round of review satisfactorily. Therefore I recommend publication of this article as it is.

Reviewer #1 (Remarks to the Author):

I have a chance to look through the revised manuscript with interest. I thank the authors to make changes with efforts. After carefully reading the manuscript, I am kind of disappointed about their arguments and knowledge. Listed below are my scientific and irrefutable criticisms:

One major concern is still the lack of novelty existing in the manuscript. The authors asked me to provide the relevant publications demonstrating this self-assembly/polymerization process. Here are two examples: *Angew. Chem. Int. Ed.* 2009, 48, 5178–5181, which demonstrated the concept by synthesizing fluorene oligomers of different lengths within the gaps and studying their transport properties; Similarly in the same journal, *Angew. Chem. Int. Ed.* 2007, 46, 3892–3895, which detailed a method to prepare multicomponent electrical circuits through the nanoscale placement of reactive groups through coordination or condensation reactions. Obviously, there are more reports related to this topic. Indeed, the concept of oligomerizing molecules evidenced in transport measurements is not new. I strongly and kindly recommend the authors to read and learn the latest advances in the field.

We would like to thank the referee for pointing our attention towards these interesting papers that deal with in-situ self-assembly. However, we are surprised that the reviewer may view those approaches as “related to this topic” in spite of the obvious differences. We note that all the studies listed by the reviewer first build the bridging molecular structure inside the junction **ex-situ** by wet synthetic chemistry (either by click chemistry in the laboratory of Chad Mirkin or by coordination chemistry, imine or amide formation in the laboratory of Collin Nuckolls), and only afterwards investigate the transport characteristics of the chemically modified junction. In contrast, in our experiments, the metalorganic oligomer is formed **in-situ** while the transport through the structure is monitored.

Specifically, the work presented in our manuscript differs from the approaches mentioned by the referee by the following essential aspects. First, we monitor the formation of the chains in *real-time*. In contrast, in the click-chemistry approach mentioned by the reviewer, the molecular assembly process cannot be followed in real-time; it is only the final result that can be assessed, namely whether the assembly process led to a junction formation or not. In our experiments, we first form a short junction (monomer) and then extend the chain *in-situ*, while recording the junction conductance during the extension process. This provides an unprecedented control over the chain extension. Second, previous studies do not have control over the length of the chain once the device electrodes have been fabricated. In our approach, the length of the chain can be decided on the fly, as the oligomerization process can be stopped at any time during the chain formation. Finally, those previous studies used a step-wise exposure of the devices to multiple solutions to trigger the chemical reactions. In our approach, only a single solution is needed throughout the whole process. The chain is being formed at the apex of the electrodes due to the presence of under-coordinated gold atoms together with the pulling of electrodes.

These novelties of our method have now been further emphasized in the manuscript.

In addition to the novelty issue, another significant issue is the unconvincing evidence for the in-situ

formation of molecular junctions as described by the authors. The authors cited a reference showing the structure with one Au as the bridge. I agree with this structure. However, they failed to provide any evidence to demonstrate two Au atoms between each BdNC unit. This is suspicious. Furthermore, this experiments seems that the Au atoms on the tip are mobile, which leaves me with an impression that the devices are unstable. The device instability is fatal and will not be useful for further applications.

First of all, we would like to stress that the limited access to the molecular structures is a well-known limitation of MCBJ measurements. This holds for measurements in all types of environments, ranging from ambient to cryogenic vacuum, and for liquid environments. Nevertheless, MCBJ is currently one of the best-established characterization tools for single molecule junctions.

Moreover, we have not claimed that we have two gold atoms bridging the isocyano compounds, as suggested by the reviewer. In the manuscript, we mention that at least 1 atom is needed, as two isocyano groups repel each other, as demonstrated by our DFT+MD simulations. Notably, those same simulations show that bridging by both 1 and 2 gold atoms is feasible. However, the concerted analysis of our electrical transport measurements and our extensive quantum electrical transport calculations strongly indicate that the isocyano compounds are actually bridged by a single gold atom. Still, we note that we do not place significant stress on this claim, because we are aware that the MCBJ technique does not provide this level of insight for a molecular junction yet.

Finally, the mobility of gold atoms at the surface of the electrodes has been known for many years in the single-molecule electronics community, and in particular the fact that atoms may even be extracted from the electrodes when molecules are strongly anchored to them. In our approach, the ductility of the Au electrodes is even an advantage, as it acts as a source of gold ions. It therefore negates the need for stepwise addition of other solutions or precursors, as commonly done for click-chemistry. Moreover, the gold atoms mobility does not prevent molecular devices from performing many different functionalities, ranging from diodes, sensors, transistors and resonant tunneling diodes, see for instance the following reviews on Molecular electronics: Aradhya *et al.* Nat. Nanotech. 2013, Xiang et al. Chem. Rev., 2016, and the focus issues in Nat. Chem. and Chem. Soc. Rev. in 2015.

Overall, the flaw and novelty issues existing in the manuscript still hamper me to recommend publication in the prestigious journal of Nature Communications. I hope that my comments help the authors to revise their manuscript for resubmission to the more specific journal.